# Modeling and Analysis of the Fractional-Order Flyback Converter in Continuous Conduction Mode by Caputo Fractional Calculus

**Chen Yang**[ID]**, Fan Xie \*, Yanfeng Chen**[ID]**, Wenxun Xiao and Bo Zhang**

School of Electric Power, South China University of Technology, Guangzhou 510000, China;
201821014509@mail.scut.edu.cn (C.Y.); eeyfchen@scut.edu.cn (Y.C.); xiaowx@scut.edu.cn (W.X.);
epbzhang@scut.edu.cn (B.Z.)
**\*** Correspondence: epfxie@scut.edu.cn; Tel.: +86-137-6065-3651

**Abstract:** In order to obtain more realistic characteristics of the converter, a fractional-order inductor and capacitor are used in the modeling of power electronic converters. However, few researches focus on power electronic converters with a fractional-order mutual inductance. This paper introduces a fractional-order flyback converter with a fractional-order mutual inductance and a fractional-order capacitor. The equivalent circuit model of the fractional-order mutual inductance is derived. Then, the state-space average model of the fractional-order flyback converter in continuous conduction mode (CCM) are established. Moreover, direct current (DC) analysis and alternating current (AC) analysis are performed under the Caputo fractional definition. Theoretical analysis shows that the orders have an important influence on the ripple, the CCM operating condition and transfer functions. Finally, the results of circuit simulation and numerical calculation are compared to verify the correctness of the theoretical analysis and the validity of the model. The simulation results show that the fractional-order flyback converter exhibits smaller overshoot, shorter setting time and higher design freedom compared with the integer-order flyback converter.

**Keywords:** fractional calculus; fractional-order flyback converter; fractional-order mutual inductance; state-space average modeling; continuous conduction mode

## 1. Introduction

In nature, many materials, phenomena, and complex processes exhibit fractional characteristics. It has been proved that fractional calculus is more accurate than integer calculus in the modeling of dynamic processes [1]. Moreover, the additional parameter orders of fractional calculus may change performance and increase design degrees of freedom for the system [2]. Therefore, fractional calculus has attracted increasing attention from the fields of engineering in recent decades, including electrochemistry, bioengineering, image processing, control theory, electrical engineering, etc [1–7].

In the field of electrical engineering, researches show that some electrical components, such as the inductor, the capacitor and the mutual inductance circuit are essentially fractional order, so it is necessary to establish fractional-order models of these components to accurately describe their actual characteristics. In [8] Jonscher et al. pointed out that there is no integer-order capacitor because the dielectric material exhibits fractional characteristics, and the impedance of an integer-order capacitor $Z(j\omega) = 1/(j\omega C)$ will violate causality. In [9], based on Curie's empirical law, Westerlund et al. proposed a new linear capacitor impedance model $Z(j\omega) = 1/[(j\omega)\alpha C]$, which can solve some problems that traditional theory cannot solve. The fractional-order capacitors with different orders and the high-power fractional-order capacitors were realized [10,11]. Moreover, the inductor was also

pointed out to be fractional order [12], and the implementation of a fractional-order inductor was discussed based on the skin effect [13]. In [14] Soltan et al. proposed the concept of fractional-order mutual inductance, and the analysis and design of fractional-order mutual inductance was carried out.

The accurate modeling of power electronic converters is of great significance for analyzing the performance of the converter and designing the closed-loop controller of the converter. Inductors, capacitors, and mutual inductance circuits are common components in power electronic converters. The accuracy of the models of these components has a great influence on the modeling of power electronic converters. However, the integer-order models of the components are usually used to build the power electronic converter model. When the components are approximately integer-order, the integer-order model of the converter can be used to approximate the characteristics of the converter. However, when the components clearly exhibit fractional characteristics, the integer-order model cannot reflect the true characteristics of the converter and may even produce wrong results. Therefore, in recent years, researchers have introduced the concept of fractional calculus to power electronic converters. In [15,16], the fractional-order capacitor was used in the power factor correction circuit and the in the buck-boost converter. However, they did not consider that the inductor can also be fractional. In [17–19], mathematical models and state-space average models of the fractional-order boost converter with a fractional-order inductor and capacitor in continuous conduction mode (CCM), discontinuous conduction mode (DCM) and pseudo continuous conduction mode (PCCM) were established to study the characteristics of the converter. Moreover, an improved equivalent small parameter method was studied for the fractional-order boost converter to solve the problem that is difficult to obtain for an analytical solution of the fractional-order DC-DC converter [20]. In [21,22], modeling and analysis of the fractional-order buck converter were performed in DCM and CCM, and the characteristics of the converter under different fractional definitions were compared. In [23,24], based on the circuit averaging technique, the mathematical models of the fractional-order buck-boost converter in CCM and DCM were established. The fractional-order buck-boost converter was further modeled and studied by the equivalent small parameter method and the Riemann-Liouville definition [25,26]. The modeling and analysis of fractional-order DC–DC converters such as buck, boost, and buck-boost under DCM and CCM were summarized in [27]. In [28], the fractional calculus was extended to the single-phase PWM rectifier, but the fractional-order inductor was not considered in the simulation. In [29], considering that the inductor and capacitor are fractional order, the fractional order modeling and analysis of the three-phase voltage source PWM rectifier was carried out. The fractional-order voltage source converter was also modeled, and stability and time-domain transient analysis were performed [30]. In short, the above literatures show that the fractional-order characteristics of components in power electronic converters will affect the modeling and performance of the converter.

Previous studies focused on the modeling and analysis of power electronic converters with a fractional-order inductor and capacitor. However, few researches focus on the fractional-order modeling and analysis of power electronic converters with mutual inductance. This would be more difficult because it would involve complex electromagnetic processes and the fractional-order model of mutual inductance. Therefore, the aim of this research is to solve the fractional-order modeling and analysis of power electronic converters with mutual inductance to extend the application of fractional calculus in power electronics. The main innovations of this research are as follows: (1) The equivalent circuit model of the fractional-order mutual inductance for power electronic converters is derived. (2) The fractional-order modeling and analysis of the flyback converter with a fractional-order mutual inductance and a fractional-order capacitor are carried out considering that the high-frequency transformer in the integer-order flyback converter is actually a mutual inductance.

This paper is organized as follows. In Section 2, the Caputo definition of fractional calculus and the basic models of fractional-order inductor and capacitor are introduced, then the equivalent circuit model of fractional-order mutual inductance are derived. In Section 3, the operating principle and the mathematical model of the fractional-order flyback converter are presented. In Section 4, the state-space average model of the fractional-order flyback converter is established, and direct current (DC) and

alternating current (AC) analyses are performed. In Section 5, the results of numerical calculation and circuit simulation are compared and analyzed. In Section 6, some conclusions are given.

## 2. Fundamentals

### 2.1. Fractional Calculus

The Caputo fractional definition is more widely used in the engineering field because the derivative of the initial value involved in the Laplace transform expression under the definition is integer order, which has a clear physical meaning. Therefore, the Caputo fractional definition is adopted in this research. The fractional derivative of the Caputo definition from [31] is

$$_{t_0}D_t^{\alpha}f(t) = \frac{1}{\Gamma(n-\alpha)}\int_{t_0}^{t}\frac{f^n(\tau)}{(t-\tau)^{1+\alpha-n}}d\tau, \tag{1}$$

where $\Gamma(\ )$ is the Gamma function, $n$ is the smallest integer greater than $\alpha$, and $\alpha$ is the order $n-1 \leq \alpha < n$. When $f(t)$ is constant, then $_{t_0}D_t^{\alpha}f(t) = 0$. The fractional integral of the Caputo definition from [31] is

$$_{t_0}D_t^{-\alpha}f(t) = \frac{1}{\Gamma(\alpha)}\int_{t_0}^{t}\frac{f(\tau)}{(t-\tau)^{1-\alpha}}d\tau, \tag{2}$$

therefore, when $f(t)$ is constant C, then

$$_{t_0}D_t^{-\alpha}f(t) = \frac{C(t-t_0)^{\alpha}}{\alpha\Gamma(\alpha)}. \tag{3}$$

The Laplace transform of the Caputo fractional derivative under zero initial conditions is

$$L\left[D_t^{\alpha}f(t)\right] = s^{\alpha}F(s). \tag{4}$$

### 2.2. The Basic Model of the Fractional-Order Inductor and Capacitor

In order to distinguish from an integer-order inductor and capacitor, the symbols of the fractional-order inductor with order $\alpha$ and the fractional-order capacitor with order $\beta$ are $L_{\alpha}$ and $C_{\beta}$, respectively. In this research, $0 < \alpha$ and $\beta \leq 1$ will be discussed. The relationships between voltage and current for the fractional-order inductor and capacitor can be expressed as

$$\begin{aligned} u_L(t) &= L_{\alpha}\frac{d^{\alpha}i_L}{dt^{\alpha}} \\ i_C(t) &= C_{\beta}\frac{d^{\beta}u_C}{dt^{\beta}} \end{aligned} \tag{5}$$

where $i_L$, $i_C$ are the currents of the fractional-order inductor and capacitor, respectively. The terms $u_L$, $u_C$ are the voltages of the fractional-order inductor and capacitor, respectively [32].

### 2.3. The Equivalent Circuit of the Fractional-Order Mutual Inductance

The symmetrical fractional-order mutual inductance model is shown in Figure 1a. Here, the primary inductance $L_{1\alpha}$, secondary inductance $L_{2\alpha}$, and mutual inductance $M_{\alpha}$ are fractional, and the orders are considered as $\alpha$. The port characteristic of the fractional-order mutual inductance can be obtained [14]:

$$\begin{aligned} u_1 &= L_{1\alpha}\frac{d^{\alpha}i_1}{dt^{\alpha}} + M_{\alpha}\frac{d^{\alpha}i_2}{dt^{\alpha}} \\ u_2 &= M_{\alpha}\frac{d^{\alpha}i_1}{dt^{\alpha}} + L_{2\alpha}\frac{d^{\alpha}i_2}{dt^{\alpha}} \end{aligned} \tag{6}$$

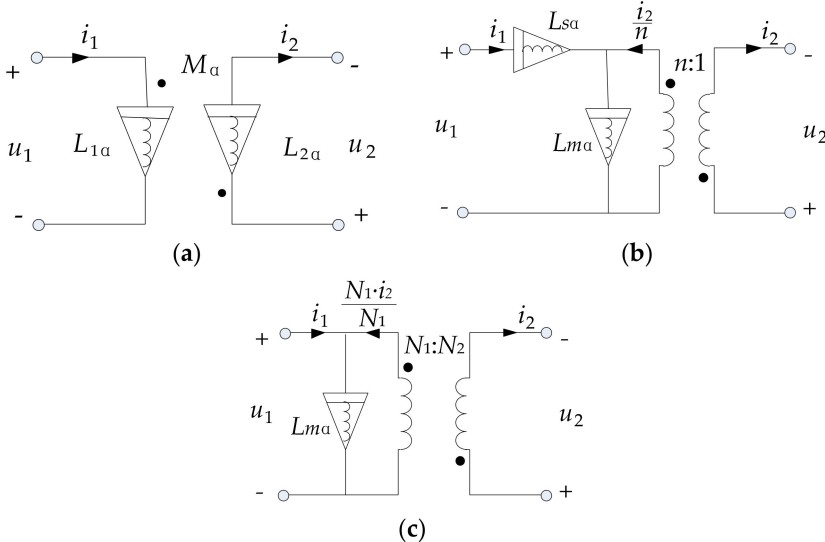

**Figure 1.** (**a**) The fractional-order mutual inductance model. (**b**) The equivalent circuit of the fractional-order mutual inductance model when the leakage inductance is considered. (**c**) The equivalent circuit of the fractional-order mutual inductance under the fully coupled condition.

When the leakage inductance is considered, the equivalent circuit of the fractional-order mutual inductance is shown in Figure 1b, and the port characteristic is

$$
\begin{aligned}
u_1 &= (L_{s\alpha} + L_{m\alpha})\frac{d^\alpha i_1}{dt^\alpha} + \frac{L_{m\alpha}}{n}\frac{d^\alpha i_2}{dt^\alpha} \\
u_2 &= \frac{L_{m\alpha}}{n}\frac{d^\alpha i_1}{dt^\alpha} + \frac{L_{m\alpha}}{n^2}\frac{d^\alpha i_2}{dt^\alpha}
\end{aligned}
\tag{7}
$$

where $L_{s\alpha}$ is leakage inductance, and the order is also $\alpha$.

When $L_{s\alpha} + L_{m\alpha} = L_{1\alpha}$, $\frac{L_{m\alpha}}{n} = M_\alpha$ and $\frac{L_{m\alpha}}{n^2} = L_{2\alpha}$, the above two models are completely equivalent. Furthermore, we can derive that $L_{s\alpha} = L_{1\alpha} - \frac{M_\alpha}{L_{2\alpha}}$, $L_{m\alpha} = \frac{M_\alpha{}^2}{L_{2\alpha}}$ and $n = \frac{M_\alpha}{L_{2\alpha}}$. Under the fully coupled condition, $M_\alpha = \sqrt{L_{1\alpha}L_{2\alpha}}$ and $\sqrt{\frac{L_{1\alpha}}{L_{2\alpha}}} = \frac{N_1}{N_2}$, and we can then obtain that $L_{s\alpha} = 0$, $L_{m\alpha} = L_{1\alpha}$ and $n = \frac{N_1}{N_2}$. The primary windings turns and secondary windings turns of the fractional-order mutual inductance are $N_1$ and $N_2$, respectively. The equivalent circuit of the fractional-order mutual inductance under the fully coupled condition is shown in Figure 1c.

## 3. Mathematical Model of the Fractional-Order Flyback Converter

Based on the integer-order flyback converter, the fractional-order flyback converter can be obtained by replacing the transformer and capacitor with the fractional-order mutual inductance and the fractional-order capacitor. The fractional mutual inductance equivalent circuit model will be used for circuit analysis and simulation. In practice, the leakage inductance is very small, and the charging and discharging time of the leakage inductance and the junction capacitor is very short compared with the main operating mode of the converter. In addition, this paper focuses mainly on the effect of the extra parameter orders on the fractional-order flyback converter when components exhibit fractional characteristics. Therefore, leakage inductance is ignored in this paper, and the voltage source $u_{in}$, the diode VD and the switch VT are considered as ideal devices. The circuit model of the fractional-order flyback converter is shown in Figure 2.

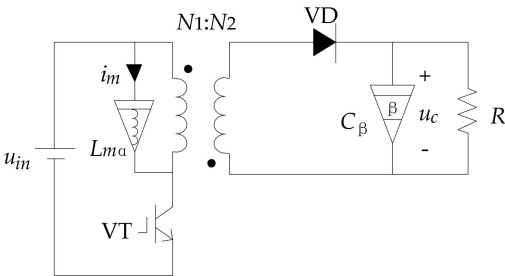

**Figure 2.** The circuit model of the fractional-order flyback converter.

When the converter operates in CCM, two main operating modes will be presented. Taking the inductor current $i_m$ and the capacitor voltage $u_c$ as state variables, the mathematical model of the fractional-order flyback converter in two operating modes are expressed.

Mode 1: The switch VT is on and the diode VD is off, for $0 \leq t < dT$. The input voltage is applied to the primary side, the transformer stores magnetic energy, and the capacitor supplies power to the load. The mathematical model in this mode is

$$\begin{bmatrix} \frac{d^\alpha i_m}{dt^\alpha} \\ \frac{d^\beta u_c}{dt^\beta} \end{bmatrix} = \begin{bmatrix} 0 & 0 \\ 0 & -\frac{1}{RC_\beta} \end{bmatrix} \begin{bmatrix} i_m \\ u_c \end{bmatrix} + \begin{bmatrix} \frac{1}{L_{m\alpha}} \\ 0 \end{bmatrix} u_{in}. \tag{8}$$

Mode 2: The switch VT is off and the diode VD is on, for $dT \leq t < T$. The magnetic energy stored in the transformer is transferred to the capacitor and the load through VD. The mathematical model in this mode is

$$\begin{bmatrix} \frac{d^\alpha i_m}{dt^\alpha} \\ \frac{d^\beta u_c}{dt^\beta} \end{bmatrix} = \begin{bmatrix} 0 & -\frac{N_1}{N_2 L_{m\alpha}} \\ \frac{N_1}{N_2 C_\beta} & -\frac{1}{RC_\beta} \end{bmatrix} \begin{bmatrix} i_m \\ u_c \end{bmatrix} + \begin{bmatrix} 0 \\ 0 \end{bmatrix} u_{in}. \tag{9}$$

## 4. State-Space Average Modeling and Analysis of the Fractional-Order Flyback Converter

### 4.1. State-Space Average Modeling and Analysis of the Fractional-Order Flyback Converter

In order to obtain the state-space average model of the fractional-order flyback converter, the variables need to be averaged. The average value $\langle x(t) \rangle$ of the variable $x(t)$ in a period is

$$\langle x(t) \rangle = \frac{1}{T} \int_{t-T}^{t} x(\tau) d\tau. \tag{10}$$

According to the properties of fractional calculus, the average value of the variable $\frac{d^\alpha x(t)}{dt^\alpha}$ in a period is

$$\begin{aligned} \left\langle \frac{d^\alpha x(t)}{dt^\alpha} \right\rangle &= \frac{1}{T} \int_{t-T}^{t} \frac{d^\alpha x(\tau)}{dt^\alpha} d\tau \\ &= \frac{d^\alpha}{dt^\alpha} \left( \frac{1}{T} \int_{t-T}^{t} x(\tau) d\tau \right) = \frac{d^\alpha \langle x(t) \rangle}{dt^\alpha} \end{aligned} \tag{11}$$

The state-space average model of the fractional-order flyback converter in CCM is

$$\begin{bmatrix} \frac{d^\alpha \langle i_m \rangle}{dt^\alpha} \\ \frac{d^\beta \langle u_c \rangle}{dt^\beta} \end{bmatrix} = \begin{bmatrix} 0 & -\frac{N_1(1-d)}{N_2 L_{m\alpha}} \\ \frac{N_1(1-d)}{N_2 C_\beta} & -\frac{1}{RC_\beta} \end{bmatrix} \begin{bmatrix} \langle i_m \rangle \\ \langle u_c \rangle \end{bmatrix} + \begin{bmatrix} \frac{d}{L_{m\alpha}} \\ 0 \end{bmatrix} \langle u_{in} \rangle \tag{12}$$

　　Assuming that a disturbance occurs near the operating point, the average values of the inductor current $i_m$, the capacitor voltage $u_c$, the duty ratio $d$, and the input voltage $u_{in}$ are expressed as the sum of the DC component and the AC component.

$$
\begin{aligned}
\langle i_m \rangle &= I_m + \hat{i}_m \\
\langle u_c \rangle &= U_c + \hat{u}_c \\
\langle u_{in} \rangle &= U_{in} + \hat{u}_{in} \\
d &= D + \hat{d}
\end{aligned}
\tag{13}
$$

　　By substituting Equation (13) into Equation (12), the following is obtained:

$$
\begin{bmatrix}
\frac{d^\alpha (I_m + \hat{i}_m)}{dt^\alpha} \\
\frac{d^\beta (U_c + \hat{u}_c)}{dt^\beta}
\end{bmatrix}
=
\begin{bmatrix}
0 & -\frac{N_1(1-D-\hat{d})}{N_2 L_{m\alpha}} \\
\frac{N_1(1-D-\hat{d})}{N_2 C_\beta} & -\frac{1}{RC_\beta}
\end{bmatrix}
\begin{bmatrix}
I_m + \hat{i}_m \\
U_c + \hat{u}_c
\end{bmatrix}
+
\begin{bmatrix}
\frac{D+\hat{d}}{L_{m\alpha}} \\
0
\end{bmatrix}
(U_{in} + \hat{u}_{in}).
\tag{14}
$$

### 4.2. DC Analysis

　　According to Equation (14), the DC component can be separated.

$$
\begin{bmatrix}
\frac{d^\alpha I_m}{dt^\alpha} \\
\frac{d^\beta U_c}{dt^\beta}
\end{bmatrix}
=
\begin{bmatrix}
0 & -\frac{N_1(1-D)}{N_2 L_{m\alpha}} \\
\frac{N_1(1-D)}{N_2 C_\beta} & -\frac{1}{RC_\beta}
\end{bmatrix}
\begin{bmatrix}
I_m \\
U_c
\end{bmatrix}
+
\begin{bmatrix}
\frac{D}{L_{m\alpha}} \\
0
\end{bmatrix}
U_{in}
\tag{15}
$$

　　Based on the fractional derivative of the Caputo definition, the quiescent operation point of the fractional-order flyback converter can be derived as

$$
\begin{aligned}
U_c &= \frac{N_2 D}{N_1(1-D)} U_{in} \\
I_m &= \frac{N_2^2 D}{N_1^2 (1-D)^2 R} U_{in}
\end{aligned}
\tag{16}
$$

　　It can be found that the quiescent operation point of the fractional-order flyback converter under the Caputo definition is only related to the circuit parameters; it is the same as the integer-order flyback converter. Let $u_{in}$ = 20 V, $D$ = 0.5, $L_{m\alpha}$ = 1 mH/s$^{1-\alpha}$, $C_\beta$ = 100 μF/s$^{1-\beta}$, $R$ = 10 Ω, $N_1$ = 50, $N_2$ = 25, and $f$ = 20 kHz; then the quiescent operation point is $U_c$ = 10 V, $I_m$ = 1 A.

　　According to Equations (3) and (8), the inductor current ripple can be deduced as

$$
\Delta i_m = \frac{U_{in}(DT)^\alpha}{\Gamma(\alpha)\alpha L_{m\alpha}}.
\tag{17}
$$

　　The inductor current ripple is not only affected by the duty ratio $D$, period $T$, input voltage $U_{in}$ and fractional-order inductor $L_{m\alpha}$, but it is also affected by the order $\alpha$ of the fractional-order inductor.

　　The condition for the fractional-order flyback converter to operate in CCM is

$$
\frac{1}{2}\Delta i_m < I_m.
\tag{18}
$$

　　According to Equations (16)–(18), the CCM operating condition can be further derived as

$$
R < \frac{2\alpha\Gamma(\alpha)DL_{m\alpha}N_2^2}{(DT)^\alpha (1-D)^2 N_1^2}.
\tag{19}
$$

　　According to Equation (19), the CCM operating area related to the order can be drawn in Figure 3. The blue area in this figure represents CCM, which means that when $\alpha$ > 0.874, the fractional-order flyback converter works in CCM.

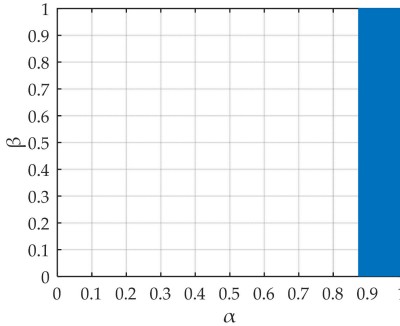

**Figure 3.** The operating area of the fractional-order flyback converter.

By using the Adomian method of fractional differential equations [33], according to Equation (8), the output voltage reduction is

$$\Delta u_C = (1 - E_\beta(-\frac{(DT)^\beta}{RC_\beta}))U_{OT}, \tag{20}$$

where $U_{OT}$ is the instantaneous value of the output voltage when VT is on, and E($\cdot$) is the Mittag-Leffler function. $U_{OT}$ can be approximated as

$$U_{OT} = U_C + \frac{1}{2}\Delta u_C. \tag{21}$$

According to Equations (16), (20), and (21), the output voltage ripple can be approximated as

$$\Delta u_C = \frac{2N_2 D U_{in}(1 - E_\beta(-\frac{(DT)^\beta}{RC_\beta}))}{N_1(1-D)(1 + E_\beta(-\frac{(DT)^\beta}{RC_\beta}))}. \tag{22}$$

It can be seen that the output voltage ripple is not only affected by the load resistance $R$, fractional-order capacitor $C_\beta$, input voltage $U_{in}$, duty ratio $D$, period $T$, primary side turns $N_1$ and secondary side turns $N_2$, but also by the order $\beta$ of the fractional-order capacitor.

The fractional-order flyback converter operates in the CCM region when $0.9 \leq \alpha$ and $\beta \leq 1$. According to Equation (17), the relationship between the inductor current ripple and the order of the fractional-order inductor is shown in Figure 4a. In addition, according to Equation (22), the relationship between the output voltage ripple and the order of the fractional-order capacitor is shown in Figure 4b. It can be seen that the inductor current ripple $\Delta i_m$ increases with the decrease of the order $\alpha$, and the output voltage ripple $\Delta u_c$ increases with the decrease of the order $\beta$.

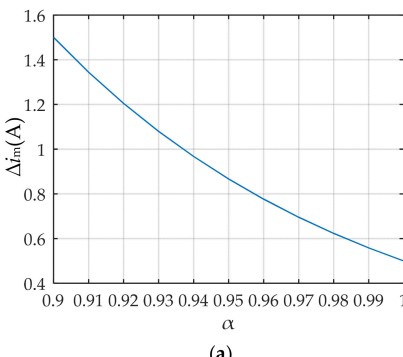

(a)

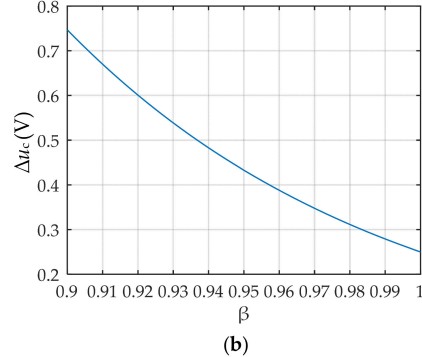

(b)

**Figure 4.** (a) The relationship between the inductor current ripple $\Delta i_m$ and the order $\alpha$. (b) The relationship between the output voltage ripple $\Delta u_c$ and the order $\beta$.

*4.3. AC Analysis*

Regardless of the quadratic terms $\hat{d}(t)\hat{u}_C(t)$, $\hat{d}(t)\hat{u}_{in}(t)$ and $\hat{d}(t)\hat{i}_m(t)$, the AC component can be obtained according to Equation (14).

$$\begin{bmatrix} \frac{d^{\alpha}\hat{i}_m}{dt^{\alpha}} \\ \frac{d^{\beta}\hat{u}_c}{dt^{\beta}} \end{bmatrix} = \begin{bmatrix} 0 & -\frac{N_1(1-D)}{N_2 L_{m\alpha}} \\ \frac{N_1(1-D)}{N_2 C_{\beta}} & -\frac{1}{RC_{\beta}} \end{bmatrix} \begin{bmatrix} \hat{i}_m \\ \hat{u}_c \end{bmatrix} + \begin{bmatrix} \frac{N_1 U_C}{N_2 L_{m\alpha}} + \frac{U_{in}}{L_{m\alpha}} \\ -\frac{N_1 I_m}{N_2 C_{\beta}} \end{bmatrix} \hat{d} + \begin{bmatrix} \frac{D}{L_{m\alpha}} \\ 0 \end{bmatrix} \hat{u}_{in} \tag{23}$$

In the frequency domain, the AC small signal model at zero initial state can be derived as

$$\begin{bmatrix} s^{\alpha}\hat{i}_m(s) \\ s^{\beta}\hat{u}_c(s) \end{bmatrix} = \begin{bmatrix} 0 & -\frac{N_1(1-D)}{N_2 L_{m\alpha}} \\ \frac{N_1(1-D)}{N_2 C_{\beta}} & -\frac{1}{RC_{\beta}} \end{bmatrix} \begin{bmatrix} \hat{i}_m(s) \\ \hat{u}_c(s) \end{bmatrix} + \begin{bmatrix} \frac{N_1 U_C}{N_2 L_{m\alpha}} + \frac{U_{in}}{L_{m\alpha}} \\ -\frac{N_1 I_m}{N_2 C_{\beta}} \end{bmatrix} \hat{d}(s) + \begin{bmatrix} \frac{D}{L_{m\alpha}} \\ 0 \end{bmatrix} \hat{u}_{in}(s). \tag{24}$$

Let $\hat{d}(s) = 0$, then the transfer function $G_{u_c u_{in}}(s)$ of the input voltage to the output voltage is

$$G_{u_c u_{in}}(s) = \left.\frac{\hat{u}_c(s)}{\hat{u}_{in}(s)}\right|_{\hat{d}(s)=0} = \frac{\frac{N_1}{N_2}(1-D)D}{L_{m\alpha}C_{\beta}s^{\alpha+\beta} + \frac{L_{m\alpha}}{R}s^{\alpha} + (1-D)^2\frac{N_1^2}{N_2^2}}. \tag{25}$$

Let $\hat{u}_{in}(s) = 0$, then the transfer function $G_{u_c d}(s)$ of the duty ratio to the output voltage can be derived as

$$G_{u_c d}(s) = \left.\frac{\hat{u}_c(s)}{\hat{d}(s)}\right|_{\hat{u}_{in}(s)=0} = \frac{\frac{N_1(1-D)}{N_2}U_{in} + \frac{N_1^2}{N_2^2}(1-D)U_C - \frac{N_1}{N_2}I_m L_{m\alpha}s^{\alpha}}{L_{m\alpha}C_{\beta}s^{\alpha+\beta} + \frac{L_{m\alpha}}{R}s^{\alpha} + (1-D)^2\frac{N_1^2}{N_2^2}}. \tag{26}$$

According to the above equations, the transfer functions are obviously related to the orders $\alpha$ and $\beta$. Considering $\alpha = 1, 0.95$ and $\beta = 1, 0.95, 0.9$, the Bode diagrams of $G_{u_c u_{in}}(s)$ and $G_{u_c d}(s)$ under different $\alpha$ and $\beta$ are shown in Figure 5. It can be seen that the amplitude–frequency characteristic curve and the phase–frequency characteristic curve are different under different orders. This means that the orders will affect the design of the closed-loop controller and the dynamic performance of the fractional-order flyback converter.

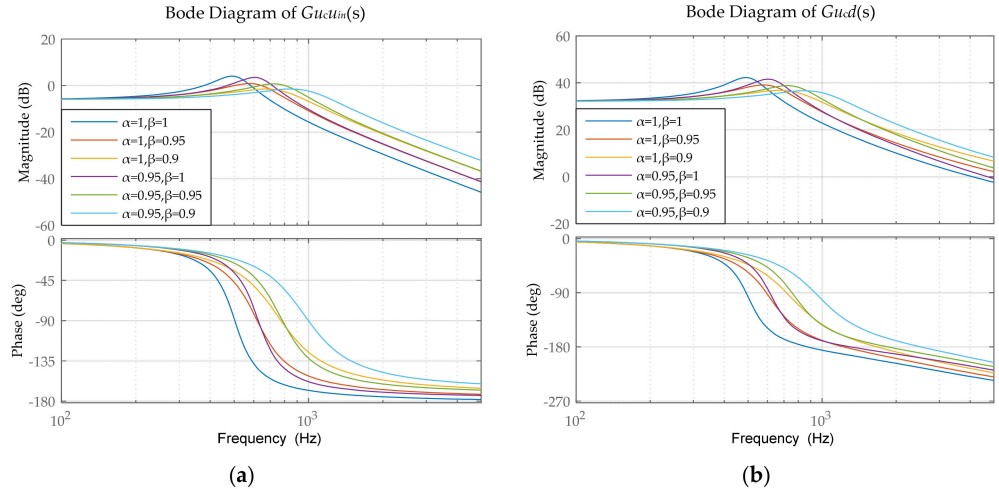

(a)　　　　　　　　　　　　(b)

**Figure 5.** (**a**) The Bode diagram of $G_{u_c u_{in}}(s)$ under different $\alpha$ and $\beta$. (**b**) The Bode diagram of $G_{u_c d}(s)$ under different $\alpha$ and $\beta$.

## 5. Simulation

### 5.1. The Rational Approximation Method for the Fractional-Order Inductor and Capacitor

Since there is currently no available fractional-order element, a method of approximating the ideal fractional-order element is used in most cases. In this research, the Oustaloup's rational approximation method is used to construct a fractional-order inductor and capacitor [34]. The circuits of the constructed fractional-order inductor and capacitor are shown in Figure 6.

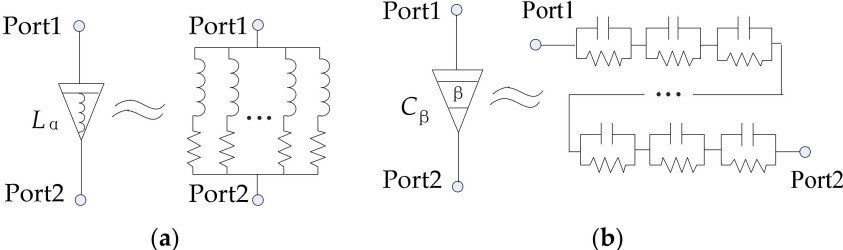

(**a**)                                                                (**b**)

**Figure 6.** (**a**) The circuit with a constructed fractional-order inductor, using Oustaloup's rational approximation method. (**b**) The circuit with a constructed fractional-order capacitor, using Oustaloup's rational approximation method.

In Oustaloup's algorithm, the lower frequency band $\omega_b = 0.01$ rad/s, the upper frequency band is $\omega_h = 10^7$ rad/s, and the filter order N = 9. The fractional-order capacitor $C_\beta = 100 \ \mu F/s^{1-\beta}$ with the orders of 0.95 and 0.9 and the fractional-order inductor $L_{m\alpha} = 1 \ mH/s^{1-\alpha}$ with the order of 0.95 are constructed and the constructed circuit parameters are shown in Table 1. The Bode diagrams of the constructed fractional-order inductor and fractional-order capacitor are shown in Figure 7. It can be seen that the circuit simulation of the amplitude–frequency characteristics of the constructed components are basically consistent with the theoretical analysis, and the phase–frequency characteristics fluctuate within an acceptable range. Therefore, the circuits of the constructed fractional-order component can be used.

**Table 1.** The circuit parameters of the fractional-order inductor and capacitor.

| $i$ | $\alpha = 0.95$ | | $\beta = 0.95$ | | $\beta = 0.9$ | |
|---|---|---|---|---|---|---|
| | $R_i$ (Ω) | $L_i$ (mH) | $R_i$ (Ω) | $C_i$ (mF) | $R_i$ (Ω) | $C_i$ (mF) |
| 1 | 5.123 k | 4.8 | 1.952 m | 0.484 | 7.337 m | 0.121 |
| 2 | 523.188 | 4.9 | 19.113 m | 0.494 | 63.945 m | 0.139 |
| 3 | 58.181 | 5.5 | 0.712 | 0.549 | 0.512 | 0.174 |
| 4 | 6.522 | 6.2 | 1.533 | 0.616 | 4.074 | 0.219 |
| 5 | 0.732 | 6.9 | 13.667 | 0.691 | 32.365 | 0.275 |
| 6 | 0.082 | 7.7 | 121.907 | 0.774 | 257.266 | 0.346 |
| 7 | 0.0091 | 8.6 | 1.095 k | 0.862 | 2.058 k | 0.433 |
| 8 | 0.943 m | 8.9 | 10.609 k | 0.890 | 17.579 k | 0.507 |
| 9 | 12.78 u | 1.2 | 0.782 M | 0.121 | 0.611 M | 0.146 |

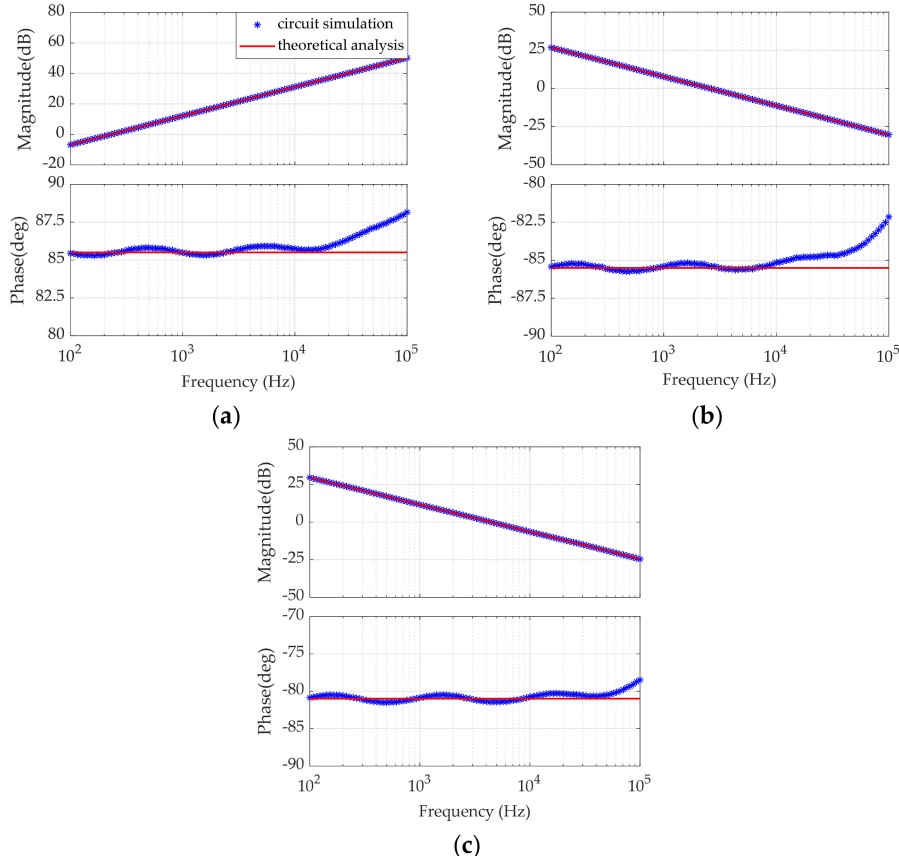

**Figure 7.** The Bode diagrams of the constructed fractional-order inductor and fractional-order capacitor. (**a**) $L_{m\alpha} = 1 \text{ mH}/s^{1-\alpha}$ and $\alpha = 0.95$. (**b**) $C_\beta = 100 \text{ μF}/s^{1-\beta}$ and $\beta = 0.95$. (**c**) $C_\beta = 100 \text{ μF}/s^{1-\beta}$ and $\beta = 0.9$.

*5.2. Simulations*

According to Figures 2 and 6, the fractional-order flyback converter circuit simulation model is built in PSIM, as shown in Figure 8. The converter is directly controlled by a PWM wave and operates in an open loop condition. The response waveforms of inductor current $i_m$ and output voltage $u_c$ under different $\alpha$ and $\beta$ are shown in Figure 9a,b, respectively. The overshoot and setting time of the output voltage under different $\alpha$ and $\beta$ are shown in Table 2. When $\alpha = 1$ and $\beta = 1$, the converter is equivalent to an integer-order flyback converter. It can be seen that the fractional-order flyback converter has the smaller overshoot and a shorter setting time than the integer-order flyback converter. When $\alpha$ is constant, the overshoot and setting time will be significantly reduced, as the capacitor order $\beta$ decreases. When $\beta$ is constant, the overshoot will be slightly reduced and the setting time will be significantly reduced as the inductor order $\alpha$ decreases. Furthermore, the capacitor order $\beta$ has a greater influence on the overshoot than that of the inductor order $\alpha$.

**Table 2.** The overshoot and setting times of the output voltage.

| $(\alpha,\beta)$ | (1,1) | (1,0.95) | (1,0.9) | (0.95,1) | (0.95,0.95) | (0.95,0.9) |
|---|---|---|---|---|---|---|
| **overshoot(V)** | 16.220 | 15.151 | 14.180 | 16.044 | 15.001 | 14.090 |
| **setting time(s)** | 0.01023 | 0.00641 | 0.00419 | 0.00743 | 0.00476 | 0.00329 |

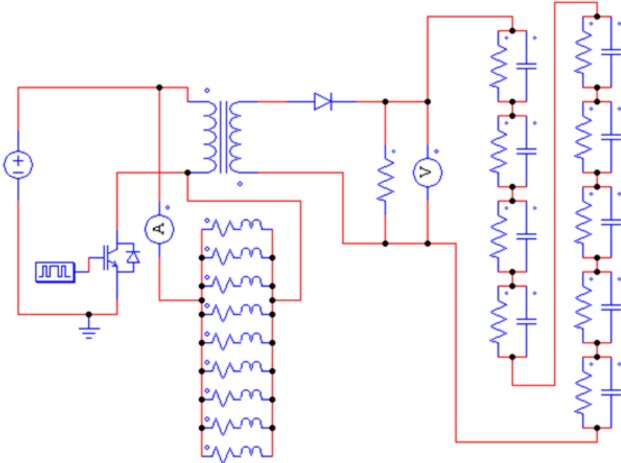

**Figure 8.** The circuit simulation model of the fractional-order flyback converter.

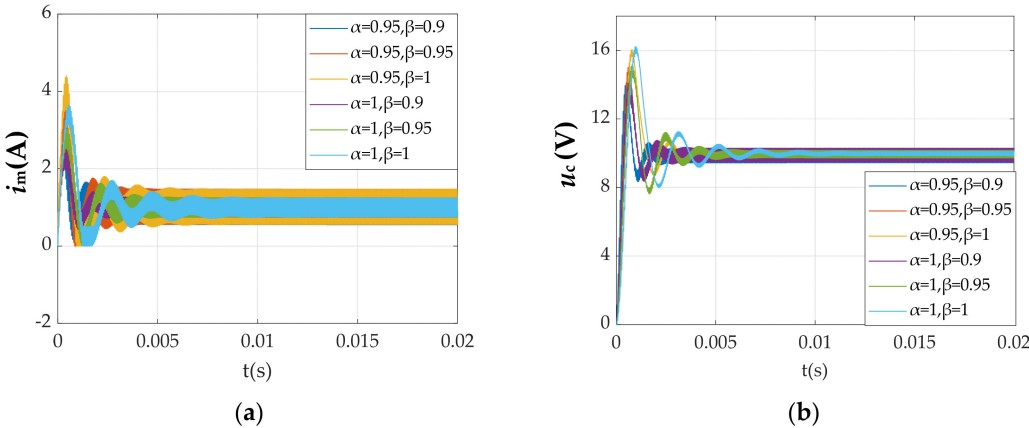

(**a**)

(**b**)

**Figure 9.** (**a**) The response waveforms of the inductor current under different α and β. (**b**) The response waveforms of the output voltage under different α and β.

In order to verify the correctness of the mathematical model of the fractional-order flyback converter established in Section 3, the mathematical model is solved based on the numerical solution method of the fractional-order extended equation in [34]. The numerical calculation results and the circuit simulation results of the inductor current $i_m$ and the output voltage $u_c$ under different α and β at steady state are shown in Figure 10. It can be seen that the numerical calculation results are basically consistent with the circuit simulation results. Therefore, the mathematical model of the fractional-order flyback converter is correct.

The circuit simulation results at the quiescent operation point, the inductor current ripple and the output voltage ripple under different α and β can be observed in Figure 10. The comparison between the circuit simulation results and the theoretical analysis results in Section 4.2 are shown in Table 3. The results show that the quiescent operation point of the fractional-order flyback converter is basically not affected by the orders. However, the inductor current ripple and capacitor voltage ripple are greatly affected by the orders. The inductor current ripple is related to the order α of the fractional-order inductor and the ripple will increase as the order decreases. Similarly, the output voltage ripple is related to the order β of the fractional-order capacitor, and the ripple will increase as the order decreases. The errors mainly come from the approximation circuits of the fractional-order element and the modeling method based on the averaging technique.

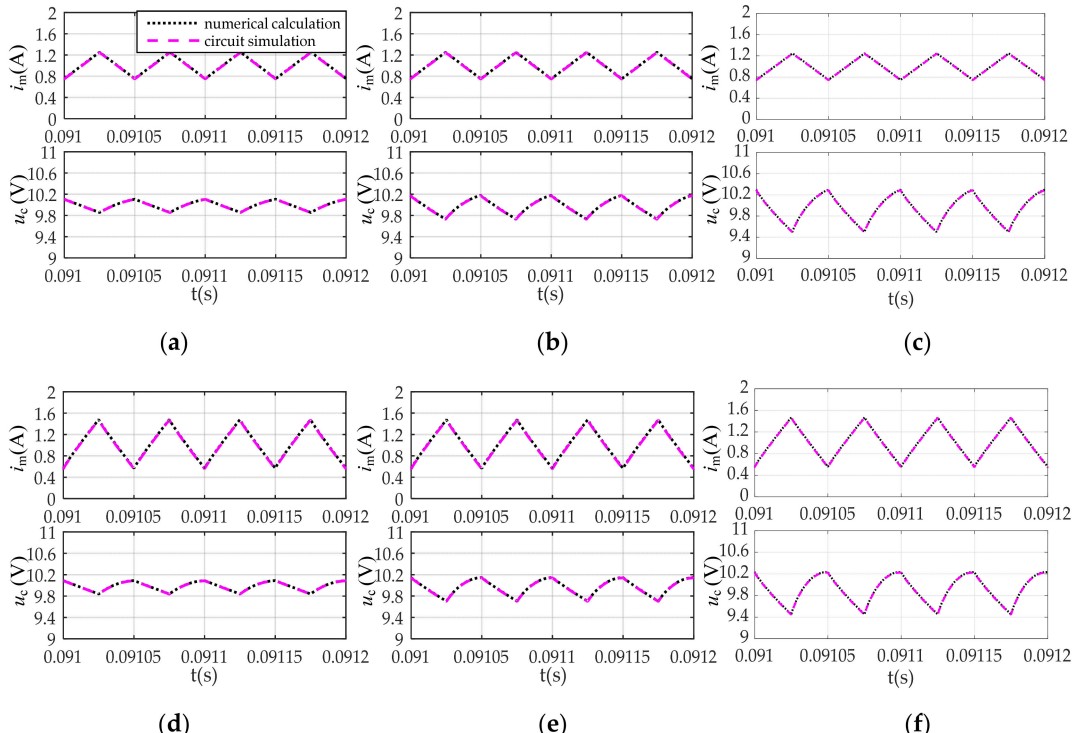

**Figure 10.** The circuit simulation and numerical calculation results under different α and β. (**a**) (α,β) = (1,1); (**b**) (α,β) = (1,0.95); (**c**) (α,β) = (1,0.9); (**d**) (α,β) = (0.95,1); (**e**) (α,β) = (0.95,0.95); and (**f**) (α,β) = (0.95,0.9).

**Table 3.** The comparison between the circuit simulation and the theoretical analysis.

| (α,β) | $U_c$ (V) | $I_m$ (A) | $\Delta i_m$ (A) | $\Delta u_c$ (V) |
|---|---|---|---|---|
| (1,1) | 9.989/10 | 0.998/1 | 0.500/0.5 | 0.249/0.25 |
| (1,0.95) | 9.970/10 | 0.996/1 | 0.500/0.5 | 0.447/0.433 |
| (1,0.9) | 9.928/10 | 0.992/1 | 0.500/0.5 | 0.796/0.747 |
| (0.95,1) | 9.980/10 | 1.015/1 | 0.911/0.867 | 0.249/0.25 |
| (0.95,0.95) | 9.955/10 | 1.013/1 | 0.914/0.867 | 0.442/0.433 |
| (0.95,0.9) | 9.901/10 | 1.006/1 | 0.914/0.867 | 0.776/0.747 |

In this research, the fractional-order inductors with α = 0.874 and α = 0.85 are also constructed. Figure 11 shows the waveforms of the inductor current when (α,β) = (0.95,1), (α,β) = (0.874,1) and (α,β) = (0.85,1). It is obvious that the converter operates in critical conduction mode when α = 0.874. However, when α > 0.874, the converter operates in CCM. When α < 0.874, the converter operates in DCM. The simulation results are consistent with the theoretical analysis in Section 4.2. For the integer-order flyback converter, the operating mode can only be changed by circuit parameters such as inductance, resistance, duty cycle and turns. However, for the fractional-order flyback converter, the operating mode can also be changed by the inductor order α. This means that the fractional-order flyback converter has better design freedom.

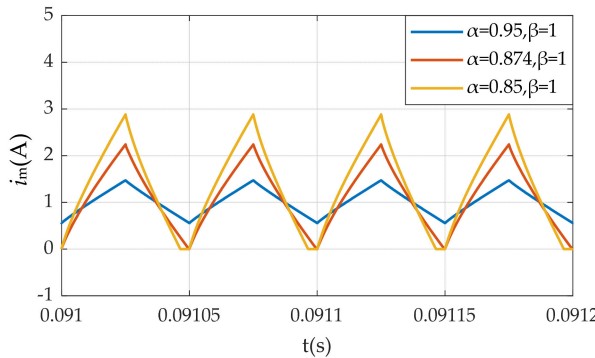

**Figure 11.** The inductor current when $(\alpha,\beta) = (0.95,1)$, $(\alpha,\beta) = (0.874,1)$ and $(\alpha,\beta) = (0.85,1)$.

The Bode diagrams of $G_{u_c u_{in}}(s)$ and $G_{u_c d}(s)$ obtained by theoretical analysis and circuit simulation under different $\alpha$ and $\beta$ are shown in Figure 12. In order to make the Bode diagrams of theoretical analysis and circuit simulation clearer, $(\alpha,\beta) = (1,1)$, $(\alpha,\beta) = (1,0.95)$ and $(\alpha,\beta) = (0.95,0.9)$ are selected. It should be noted that in the circuit simulation, the original switching model of the converter is used to obtain Bode diagrams of the transfer functions. In Figure 12, the solid line is the result of theoretical analysis, and the asterisks are the result of the circuit simulation. The circuit simulation results are basically consistent with the theoretical analysis results. Therefore, the AC analysis of the fractional-order flyback converter is correct.

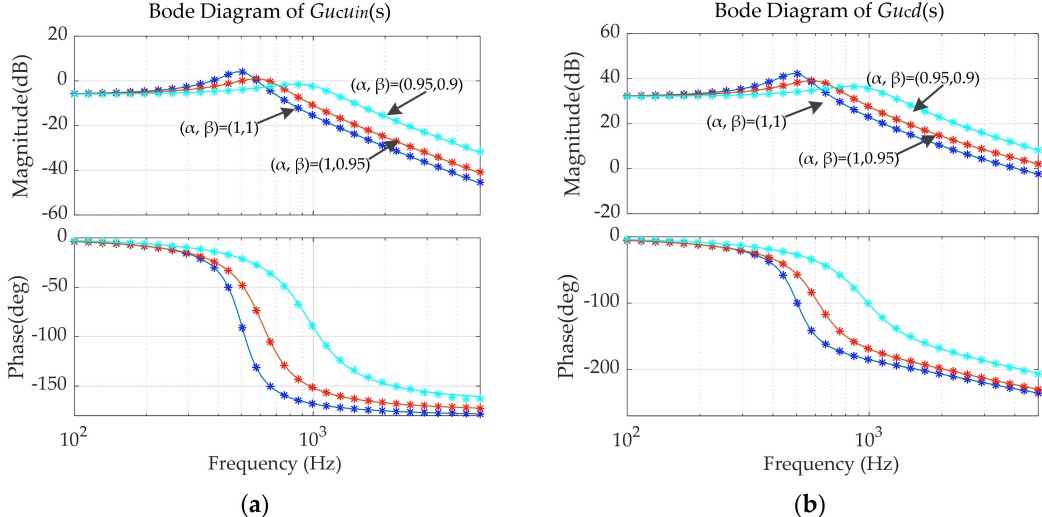

**Figure 12.** (**a**) The Bode diagram of $G_{u_c u_{in}}(s)$ obtained by theoretical analysis and circuit simulation under different $\alpha$ and $\beta$. (**b**) The Bode diagram of $G_{u_c d}(s)$ obtained by theoretical analysis and circuit simulation under different $\alpha$ and $\beta$.

## 6. Conclusions

In this study, the fractional-order mutual inductance is extended to power electronic converters. The fractional-order flyback converter with fractional mutual inductance and a fractional-order capacitor is proposed. The equivalent circuit model of the fractional-order mutual inductance is established. Based on Caputo fractional calculus, modeling and analysis for the fractional-order flyback converter in CCM are carried out. Numerical simulation and circuit simulation verify the validity of theoretical analysis and modeling.

It can be observed that the order $\alpha$ of the fractional-order inductor will affect the inductor current ripple and the CCM operating condition, and the order $\beta$ of the fractional-order capacitor will affect the output voltage ripple. This means that when the components are fractional-order, the converter

obviously shows fractional characteristics, so the fractional-order mutual inductance model should be adopted in the modeling of power electronic converters with mutual inductance. Based on the AC analysis, the transfer functions are also proved to be related to the orders α and β. This will give an indication of the design of the converter controller. In addition, compared with the integer-order flyback converter, the fractional-order flyback converter also shows better characteristics, including smaller overshoot, shorter setting time and higher degrees of freedom. Therefore, the fractional-order flyback converter can obtain better characteristics by adjusting the orders.

This work provides a reference for the application of fractional calculus in power electronic converters with mutual inductance. In addition, it may also inspire work in other fields, such as circuit theory, wireless power transfer systems, control systems and so on.

**Author Contributions:** Conceptualization, C.Y. and F.X.; methodology, C.Y. and F.X.; software, C.Y.; validation, C.Y. and F.X.; formal analysis, C.Y. and F.X.; investigation, C.Y.; data curation, C.Y.; writing—original draft preparation, C.Y.; writing—review and editing, F.X., Y.C., W.X. and B.Z.; supervision, F.X, Y.C., W.X and B.Z.; project administration, F.X.; funding acquisition, F.X. All authors have read and agreed to the published version of the manuscript.

**Funding:** This research was funded by the National Key Research and Development Program of China, grant number 2018YFB0905803.

**Conflicts of Interest:** The authors declare no conflict of interest.

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
