# Peer review of "Modeling and Analysis of the Fractional-Order Flyback Converter in Continuous Conduction Mode by Caputo Fractional Calculus"

_electronics, doi:10.3390/electronics9091544_

Round 1

Reviewer 1 Report

In my opinion, the corrections introduced by the authors are satisfactory and allow for a positive assessment of the article. In my opinion, the introduction could be prepared more legibly, but I accept the current, amended version.

Author Response

Response to Reviewer 1 Comments

Thank you very much for your review and comments on our paper. We have seriously considered about your comments and have incorporated them in the revised manuscript. In the following, we list our responses to your comments.

Point : In my opinion, the corrections introduced by the authors are satisfactory and allow for a positive assessment of the article. In my opinion, the introduction could be prepared more legibly, but I accept the current, amended version.

Response: Thank you very much for your comment and recognition. In this revised manuscript, we have modified sentences "The aim of this research is to solve the fractional-order modeling and analysis of power electronic converters with mutual inductance to extend the application of fractional calculus in power electronics. The main innovations of this research are as follows: (1) The equivalent circuit model of the fractional-order mutual inductance for power electronic converters is derived. (2) Considering that the high-frequency transformer in the integer-order flyback converter is actually a mutual inductance, the fractional-order modeling and analysis of the flyback converter with fractional-order mutual inductance and fractional-order capacitor are carried out." to more clearly clarify the aim and innovation of our research (lines 84-91).

Yours sincerely,

Chen Yang

Reviewer 2 Report

I have the following comments:

  1. Please state more clearly the aim and the novelty of your research. You may want to add two sentences starting with “the aim is ….” and “the novelty consists of … “.
  2. Please cite the origin of the equations 1-7.
  3. I would recommend you to include a case study with numerical values, which clearly shows results obtained by both methods. This would offer a better comparison and would support your approach.
  4. One of the most important parameters of the flyback transformer is the leakage inductance and the size of the air gap. Would you include it in your analysis? In this sense, the analysis of the magnetic energy stored in the transformer (lines 150-153) can be extended as well.
  5. You state that the model is correct based on numerical simulations and calculations (line 276). Such a statement would be more convictive if supported by experimental verifications.
  6. Do you need an R-C-D clamp in the primary side of the transformer, which usually is an obligatory component of an off-line flyback concreter? If YES, you may want to include it in your analysis and the final simulation, figure 7.
  7. Would you present in the same oscillogram primary and secondary voltages and currents. You may want to offer a more precise analysis of the switch-on and switch-off times.
  8. Your conclusions are that your model shows smaller overshoot and shorter setting time, obviously compared to another model (line 335). Could that result be supported with an experimental verification?
  9. Would you include in your analysis the losses, hence the expected efficiency of the converter.
  10. Having improved the previous comments, you would be able to improve the conclusions as well. It would be more explicate why a designer would prefer to use your approach rather than the conventional approaches for flyback converter design.

Thank you for the interesting paper.

Round 2

Reviewer 2 Report

After my comments were competently answered and the paper has been significantly improved, I would recommend the paper to be published in the present form.
Thank you.

This manuscript is a resubmission of an earlier submission. The following is a list of the peer review reports and author responses from that submission.

Round 1

Reviewer 1 Report

Interesting and good paper over all

Reviewer 2 Report

The added value of the content is limited. I am writing this, because important relevant references, which the same subject, are missed.

Including these references it will be obvious the weak contribution made by the authors.

See for example:

https://onlinelibrary.wiley.com/doi/abs/10.1002/cta.2840

https://europepmc.org/article/med/28709651

https://pubmed.ncbi.nlm.nih.gov/28709651/

Reviewer 3 Report

The research presented in the article is interesting and deserves the reader's attention, so there are grounds for publishing the work in a scientific journal. The article thematically meets the requirements for publication in the Electronics magazine. The presented study is scientifically important, the presented argument is quite clear, the structure of the work is correct. The article requires linguistic and editorial corrections. It is definitely necessary to broaden and correct the conclusions so that they refer to the goal, the results achieved, but also present the authors' contribution to the development of knowledge on the issue under study. Therefore, the obtained results should be related to other scientific studies, in particular those published in good and very good periodicals. A more solid literature review is missing in the presented work. 20 publications is a relatively small number, so this part of the work should be improved.

Reviewer 4 Report

This is an interesting work, the paper is well organized and the results are well presented. However, more experiments are needed to make this work publishable.  My comments are as follow
1. What is non-isolated fraction-order and isolated fraction-order? What is the advantage and disadvantage of these models? This is the main point that makes your model standing out over the approaches in [14,15]

2. How this model can be applied for DCM?

3. he comparisons show that the model provides the calculation results close to the simulation one which is a good point of this article, however it also the drawback. If the simulation can do the job and so what is the point to have such a model?

4. It is better if the model can predict more parameters of the DC-DC converter such as losses, efficiency, line regulation,...etc rather than only the voltage ripple, inductor current.

